# Anti-Pulmonary Fibrosis Activities of Triterpenoids from *Oenothera biennis*

**DOI:** 10.3390/molecules27154870

**Published:** 2022-07-29

**Authors:** Juanjuan Liu, Jingke Zhang, Mengnan Zeng, Meng Li, Shuangshuang Xie, Xiaoke Zheng, Weisheng Feng

**Affiliations:** 1College of Pharmacy, Henan University of Chinese Medicine, Zhengzhou 450046, China; l18638238805@163.com (J.L.); 18137802812@163.com (J.Z.); 17320138484@163.com (M.Z.); limeng31716@163.com (M.L.); shuangxie@hactcm.edu.cn (S.X.); zhengxk.2006@163.com (X.Z.); 2The Engineering and Technology Center for Chinese Medicine Development of Henan Province, Zhengzhou 450046, China

**Keywords:** *Oenothera biennis* L., triterpenoids, anti-pulmonary fibrosis activities, BEAS-2B cells, RTCA

## Abstract

Five new triterpenoids, oenotheralanosterols C-G (**1**–**5**), with seven known triterpenoidcompounds, namely 2*α*,3*α*,19*α*-trihydroxy-24-norurs4,12-dien-28-oic acid (**6**), 3*β*,23-dihydroxy-1-oxo-olean-12-en-28-oic acid (**7**), remangilone C (**8**), knoxivalic acid A (**9**), termichebulolide (**10**), rosasecotriterpene A (**11**), androsanortriterpene C (**12**), were extracted and separated from the dichloromethane part of *Oenothera biennis* L. The anti-pulmonary fibrosis activities of all the compounds against TGF-*β*1-induced damage tonormal human lung epithelial (BEAS-2B) cells were investigated in vitro. The results showed that compounds **1–****2**, **6**, **8**, and **11** exhibited significant anti-pulmonary fibrosis activities, with EC_50_ values ranging from 4.7 μM to 9.9 μM.

## 1. Introduction

Pulmonary fibrosis (PF) is an end-stage change of a large group of lung diseases which can lead to death if it is not properly treated [1]. The main causes of PF are the inhalation of foreign particles (such as dust and asbestos fibers), infections (such as COVID-19), autoimmune diseases (such as systemic autoimmune diseases of the connective tissue), and exposure to radiation treatment (such as radiation therapy for lung or breast cancer) [2]. Currently, PF is mainly conventionally treated using anti-inflammatory and immunosuppressant drugs (such as pirfenidone and nintedanib). However, these drugs can only retard the progression and relieve the symptoms of PF [3]. At the same time, an increasing number of Traditional Chinese Medicine therapies have been proven effective in treating PF [4,5,6,7]. Particularly, triterpenoids were proven to be bioactive structures against PF [8].

*Oenothera* is a genus that includes more than 200 species that are distributed in various regions of the world. These species can be found in the southern parts (northeast) and mountainous areas of China [9]. These species of *Oenothera biennis* are rich in a variety of natural compounds, including flavonoids, tannins, fatty acids, and terpenoids [10,11,12,13,14]. Modern pharmacological research has shown that these species possess antibacterial, anti-inflammatory, antioxidant, blood lipid lowering, blood sugar lowering, and anti-tumor properties, as well as other pharmacological effects [15,16,17,18,19,20,21]. Among these species, *Oenothera biennis* is a perennial herbaceous plant which is commonly used in folk medicine [22]. In consulting local books, we found that *O*. *biennis* can be used to treat lung-related diseases. For example, “Quanzhou Materia Medica” recorded that *O*. *biennis* can be used to treat lung disease and fistulas. Meanwhile, according to the literature, it is found that the triterpenoids in evening primrose have antiproliferative, antimicrobial efficacy, with free radical scavenging and ferric reducing activities [12,23]. In this study, we investigated and identified five new (**1**–**5**) and seven known triterpenoids (**6**–**12**) from the dichloromethane fraction of *O. biennis* (Figure 1). Furthermore, some compounds have shown significant protective effects against TGF-*β*1-induced PF in healthy human lung epithelial (BEAS-2B) cells, which suggests the potential anti-pulmonary fibrosis activities of these compounds.

## 2. Results and Discussion

Compound **1** (Figure 1) was isolated as a white amorphous powder (20.6 mg), and the molecular formula was determined as C_30_H_48_O_5_, based on the quasi-molecular ion at *m*/*z* 511.3397 [M + Na]^+^ (calculated for C_30_H_48_O_5_Na, 511.3399) in the HR-ESI-MS. The IR absorption bands confirmed the presence of -OH (3369 cm^−1^) and -COOH (1636 cm^−1^) in compound **1**. The UV spectrum exhibited an absorption band at *λ* 206 nm corresponding to the carbonyl group. The ^1^H NMR spectrum (Table 1) showed three oxygenated methines at [*δ*_H_ 4.45 (1H, m, H-6), 3.70 (1H, m, H-2), and 2.86 (1H, d, *J* = 9.5 Hz, H-3)] and seven methyl groups at [*δ*_H_ 0.77 (3H, s, H-30), 0.93 (3H, s, H-29), 1.07 (3H, s, H-23), 1.16 (3H, s, H-24), 1.17 (3H, s, H-27), 1.17 (3H, s, H-26),and 1.32 (3H, s, H-25)]. The ^13^C NMR (Table 2) and DEPT 135 spectra exhibited 29 carbon resonances, including a carboxylic carbon at (*δ*_C_ 180.6), two olefinic carbons at (*δ*_C_ 139.1 and 129.8), three oxygenated methine carbons at (*δ*_C_ 68.8, 69.8, and 84.7), and seven methyl carbons at (*δ*_C_ 32.7, 28.8, 24.7, 21.8, 19.8, 19.5, and 18.7). The HMBC and ^1^H–^1^H COSY correlations of compound **1** are shown in Figure 2. The ^1^H–^1^H COSY spectrum, including *δ*_H_ 1.61, 1.52 (H-11), *δ*_H_ 1.62 (H-9), *δ*_H_ 2.81, and 1.88 (H-12), was consistent with a double bond in ring D. Meanwhile, it can be observed that the attachment of a double bond of C-13 and C-18 was established by HMBC correlations from *δ*_H_1.17 (H_3_-27) to *δ*_C_139.1 (C-13) and from *δ*_H_ 2.27 (H-19) to *δ*_C_ 129.5 (C-18). The analysis of the^13^C NMR data of compound **1** established its close structural resemblance to 2α, 3*β*-dihydroxy-yolean-13(18)-en-28-oic acid [24]; however, the only difference was the presence ofan oxygenated methine *δ*_C_ 68.8 (C-6) in compound **1** and the absence of a methylenesignal compared to 2α, 3*β*-dihydroxy-yolean-13(18)-en-28-oic acid. The HMBC correlations from *δ*_H_ 4.45 (H-6) to *δ*_C_ 56.8 (C-5), *δ*_C_ 43.1 (C-7), and *δ*_C_ 39.3 (C-10) were observed. The relative configuration of compound **1** was determined based on a NOESY experiment. The NOESY correlation (Figure 3) was observed between H-2and H_3_-25, which indicated the *α*-orientation of the hydroxy group at C-2. The correlations of H-3 and H-5 demonstrated the *α*-orientation of H-3, which indicated that the hydroxy group at C-3 was *β*-oriented. Additionally, the NOESY correlation was observed between H-6and H_3_-23/H-5, which indicated that the hydroxy group at C-6 was *β*-oriented [25,26]. Thus, the structure of compound **1** was determined to be 2*α*,3*β*,6*β*-trihydroxy-yolean-13(18)-en-28-oic acid, and it was named oenotheralanosterol C.

Compound **2** (Figure 1) was isolated as a white amorphous powder (11.0 mg), and its molecular formula was established as C_29_H_42_O_5_ from the molecular ion peak at *m*/*z* 493.2934 [M + Na]^+^ (calculated for C_29_H_42_O_5_Na, 493.2930) in the HR-ESI-MS. The IR absorption bands confirmed the presence of -OH (3380 cm^−1^) and -COOH (1644 cm^−1^) groups. The UV spectrum exhibited an absorption band at *λ* 204 (278 nm), corresponding to the carbonyl group. The ^1^H NMR spectrum (Table 1) showed an olefinic proton at *δ*_H_ 5.28 (1H, t, *J* = 3.8 Hz) and six methyl groups [(*δ*_H_ 1.85, (3H, d, *J* = 2.0 Hz, H-23), 1.39 (3H, s, H-27), 1.14 (3H, s, H-29), 1.00 (3H, d, *J* = 6.6 Hz, H-30), 0.93 (3H, s, H-25), and 0.85 (3H, s, H-26)]. The ^13^C NMR (Table 2) and DEPT 135 spectra revealed 29 signals, with a ketocarbonyl carbon at (*δ*_C_ 195.7), a carboxylic carbon at (*δ*_C_ 182.8), two double bonds at (*δ*_C_140.0, 128.1 and 145.3, 133.1), an oxygenated quaternary carbon at (*δ*_C_ 74.2), and six methyl carbons at (*δ*_C_ 29.5, 24.5, 17.7, 16.2, 14.4, and 13.4). The HMBC and ^1^H–^1^H COSY correlations of compound **2** are illustrated in Figure 2. Meanwhile, an analysis of the ^1^H and ^13^C NMR data of compound **2** established its close structural resemblance to 2-oxo-3*J*,19*α*-dihydroxy-24-nor-urs-12-en-28-oic acid [27]. The only difference between the compound and the resemblance structure is the presence of a double bond at C-3/C-4 in compound **2**. It can be observed that the attachment of a double bond to C-3 and C-4 was established by the HMBC correlations from *δ*_H_ 2.56 and 2.16 (H-1) to *δ*_C_ 195.7 (C-2), *δ*_C_ 145.3 (C-3), *δ*_C_49.9 (C-5), and *δ*_C_39.3 (C-10) and from *δ*_H_ 1.00 (H_3_-23) to *δ*_C_ 145.3 (C-3), *δ*_C_133.1(C-4), and *δ*_C_49.9 (C-5). The relative configuration of compound **2** was determined based on a NOESY experiment. The NOESY correlations (Figure 3) of H_3_-29/H-12/H_3_-26 showed the *J*-orientation of H_3_-29, which indicated that the hydroxy group at C-29 was *α*-oriented [28]. Thus, compound **2’s** structure was determined to be 2-oxo-3,19*α*-dihydroxy-24-nor-urs-12-en-28-oic acid, and it was named oenotheralanosterol D. The data are available in Appendix A.

Compound **3** (Figure 1) was isolated as a white amorphous powder (12.8 mg), possessing the molecular formula of C_30_H_46_O_5_, based on HR-ESI-MS at *m*/*z* 509.3237 [M + Na]^+^ (calculated for C_30_H_46_O_5_Na, 509.3243). The IR absorption bands confirmed the presence of -OH (3368 cm^−1^) and -COOH (1686 cm^−1^) groups. The UV spectrum showed an absorption bandat *λ* 204 nm, corresponding to the carbonyl group. The ^1^H NMR spectrum (Table 1) showed an olefinic proton at *δ*_H_ 5.27 (1H, t, *J* = 3.6 Hz, H-12), an oxygenated methine at *δ*_H_ 3.36 (1H, dd, *J* = 4.6, 12.2 Hz, H-3), and seven methyl groups [*δ*_H_ 0.86 (3H, s, H-26), 0.94 (3H, d, *J* = 6.7 Hz, H-30), 1.01 (3H, s, H-23), 1.04 (3H, s, H-24), 1.22 (3H, s, H-29), 1.32 (3H, s, H-25), and 1.36 (3H, s, H-27)]. The ^13^C NMR (Table 2) and DEPT 135 spectra revealed 30 signals, with a ketocarbonyl carbon at (*δ*_C_ 215.3), a carboxylic carbon at (*δ*_C_ 182.2), a double bond at (*δ*_C_139.4 and 130.0), an oxygenated quaternary carbon at (*δ*_C_ 73.5), an oxygenated methine carbon at (*δ*_C_ 79.7), and seven methyl carbons at (*δ*_C_ 29.0, 27.0, 24.8, 18.1, 16.6, 16.6, and 15.9). Furthermore, an analysis of the^13^C NMR data of compound **3** established its close structural resemblance to 2-oxo-pomolic acid [29], which only differed in the position of the ketocarbonyl (*δ*_C_ 215.3, C-6) and replaced (*δ*_C_ 215.3, C-1) in the compound. The HMBC correlations from *δ*_H_ 3.10, 2.26 (H-2) to *δ*_C_ 215.3 (C-1), *δ*_C_ 79.7 (C-3), and *δ*_C_40.4 (C-4) were observed. The relative configuration of compound **3** was determined based on a NOESY experiment. The NOESY correlations (Figure 3) of H-3/H_3_-23/H-5 demonstrated that the H-3 and H_3_-23 were *α*-oriented, indicating the *J*-orientation of the hydroxy group at C-3. The correlations of H_3_-29/H-12/H_3_-26 demonstrated that H_3_-29 was *J*-oriented, indicating the *α*-orientation of the hydroxy group at C-29 [30,31]. Thus, the structure of compound **3** was determined to be 3*J*,19*α*-dihydroxy-1-oxo-olean-12-en-28-oic acid, and it was named oenotheralanosterol E.

Compound **4** (Figure 1) was isolated as a white amorphous powder (11.7 mg), and itsmolecular formula was determined to be C_30_H_46_O_6_, based on the quasi-molecular ion at *m*/*z* 525.3180 [M + Na]^+^ (calculated for C_30_H_46_O_6_Na, 525.3192) in the HR-ESI-MS analysis. The IR absorption bands confirmed the presence of -OH (3366 cm^−1^) and -COOH (1688 cm^−1^) groups. The UV spectrum showed an absorption band at *λ* 203 nm, corresponding to the carbonyl group. The ^1^H NMR (Table 1) and ^13^C NMR (Table 2) data of compound **4** were similar to those of compound **3**, differing only in the absence of a methyl group, which was replaced by an oxygenated methylene carbon (*δ*_C_ 66.0, C-23) in compound **4**. The HMBC and ^1^H–^1^H COSY correlations of compound **4** are illustrated in Figure 2. The HMBC correlations from *δ*_H_ 3.51, 3.35 (H_3_-23) to *δ*_C_ 79.7 (C-3), *δ*_C_ 40.4(C-4), *δ*_C_ 55.8 (C-5), and *δ*_C_ 13.3 (C-24) were observed. The relative configuration of compound **4** was determined based on a NOESY experiment. The NOESY correlations (Figure 3) of H-3/H_3_-23/H-5 demonstrated that H-3 and H_3_-23 were *α*-oriented, implying the *J*-orientation of the hydroxy group at C-3. The correlations of H_3_-29/H-12/H_3_-26 demonstrated the *J*-orientation of H_3_-29, indicating the *α*-orientation of the hydroxy group at C-29 [32]. Thus, the structure of compound **4** was determined to be 3*J*,19*α*-dihydroxy-1-oxo-olean-12-en-23-ol-28-oic acid, and it was named oenotheralanosterol F.

Compound **5** (Figure 1) was isolated as a white amorphous powder (3.4 mg), and its molecular formula was established as C_28_H_42_O_7_ from the molecular ion peak at *m*/*z* 513.2832 [M + Na]^+^ (calculated for C_28_H_42_O_7_Na, 513.2828) in the HR-ESI-MS. The IR absorption bands confirmed the presence of -OH (3371, 2951 cm^−1^) and -COOH (1650, 1388 cm^−1^) groups. The UV spectrum exhibited an absorption band at *λ* 204 nm, corresponding to the carbonyl group. The ^1^H NMR spectrum (Table 1) showed an olefinic bond at *δ*_H_ 5.35 (1H, t, *J* = 3.8 Hz, H-12), two oxygenated methines at [*δ*_H_ 3.22 (1H, d, *J* = 3.8 Hz, H-22) and 3.91 (1H, dd, *J* = 4.4, 11.6 Hz, H-19)], and six methyl groups at [*δ*_H_ 0.84 (3H, s, H-26), 0.97 (3H, s, H-29), 1.03 (3H, s, H-30), 1.09 (3H, s, H-25), 1.34 (3H, s, H-27), and 2.23 (3H, s, H-23)]. The ^13^C NMR (Table 2) and DEPT 135 spectra revealed 28 signals, with a ketocarbonyl carbon at (*δ*_C_ 215.3), two carboxylic carbons at (*δ*_C_ 180.7 and 175.2), a double bond at (*δ*_C_143.8 and 125.1), two oxygenated methine carbons at (*δ*_C_ 72.4 and 82.0), and six methyl carbons at (*δ*_C_ 31.6, 28.8, 26.2, 25.2, 18.4, and 17.5). The HMBC and ^1^H–^1^H COSY correlations of compound **1** are shown in Figure 2. Furthermore, an analysis of the^1^H and ^13^C NMR data of compound **5** established its close structural resemblance toivorengenin B [33], and the only difference was that the oxygenated methine (*δ*_C_ 72.4, C-22) replaced the methylene signal (*δ*_C_ 32.5, C-22). The HMBC correlations from *δ*_H_ 3.91 (H-22) to *δ*_C_ 53.0 (C-17), *δ*_C_ 182.2 (C-28), and *δ*_C_38.0 (C-21) were observed. The relative configuration of compound **5** was determined based on a NOESY experiment. The NOESY correlations (see Figure 3) of H-22/H-18/H_3_-27 showed that H-18 and H-22 were *a*-oriented, indicating the *J*-orientation of the hydroxy group at C-22. The correlations of H-19/H-12/H_3_-26 showed that H-19 was *J*-oriented, indicating the *a*-orientation of the hydroxy group at C-19 [34]. Thus, the structure of compound 5 was determined to be 19*a*,22*J*-dihydroxy-4-oxo-3,24-dinor-2,4-secoolean-12-ene-2,28-dioic acid, and it was named oenotheralanosterol G.

In addition, the known compounds **6**–**12** were also obtained from the dichloromethane part of *O**. biennis.* Comparing the NMR spectroscopic data of compounds **6**–**12** with the reported literature, the known compounds were identified as 2*α*,3*α*,19*α*-trihydroxy-24-norurs4(23),12-dien-28-oic acid (**6**) [35], 3*J*,23-dihydroxy-1-oxo-olean-12-en-28-oic acid (**7**) [36], remangilone C (**8**) [37], knoxivalic acid A (**9**) [38], termichebulolide(**10**) [39], rosasecotriterpene A (**11**) [40], and rosanortriterpene C (**12**) [41], respectively.

The TGF-*J*1-induced lung slice fibrosis model provided an experimental basis for the study of the pathological mechanism of PF and therapeutic drugs. In this paper, we used this model to explore the anti-pulmonary fibrosis activities of the isolated compounds through real-time cell analysis. The results indicated that compounds **1**–**2**, **6**, **8**, and **11** significantly decreased the damage of BEAS-2B cells induced by TGF-*J*1, with EC_50_ values ranging from 4.7 μM to 9.9 μM (Table 3). It is speculated that these compounds may have potential activities against PF.

According to our experimental results, the lung-protective activities are influenced by multiple structural factors. For example, compound **6** exhibited the most effective anti-pulmonary fibrosis activities compared to the other compounds; therefore, the double bond at C-4/C-23 might be an active group that could increase the structures’ activities. Similarly, compound **10** showed weaker anti-pulmonary fibrosis activities compared to the other compounds, which means the esterification of the carboxyl group at C-28 could decrease the compounds’ activities.

## 3. Materials and Methods

### 3.1. General Procedures

The NMR spectra were recorded using a Bruker AVANCE III 500 nuclear magnetic resonance instrument and mass spectra was completed with a Bruker maxis HD time-of-flight mass spectrometer (Bruker, Germany). The UV and IR spectra were recorded on a Thermo EVO 300 spectrometer (Thermo, Waltham, MA, USA) and a Thermo Nicolet IS 10 spectrometer (Thermo, Waltham, MA, USA). The separation of compounds was achieved on an LC-52 HPLC (separation Beijing Technology, SP-5030 semi-preparative high-pressure infusion pump, UV200 detector, easy Chromchromatographic workstation, with a COSMOSIL C18-MS-II chromatographic column of 250 mm × 20 mm, 5 μm). A Multiskan MK3 microplate reader (Thermo Fisher, Waltham, MA, USA) was used in the bioassay, along with a carbon dioxide type 3111 incubator (Thermo, Waltham, MA, USA) and a Centrifuge-5804R high speed centrifuge (Eppendorf, Germany). Column chromatography (CC) was performed using an MCI gel CHP-20 (TOSOH Corp, Tokyo, Japan), a Sephadex LH-20 (40–70 mm, anAmersham Pharmacia Biotech AB, Uppsala, Sweden) and silica gel (200–300 mesh, Marine Chemical Industry, Qingdao, China). The chemical reagents were supplied by the Beijing Chemical Plant (Beijing, China), and the RTCA from Agilent, Santa Clara, CA, USA and TGF-*J*1 from PeproTech, Cranbury, NJ, USA.

### 3.2. Plant Material

The *Oenothera biennis* L. specimens were collected in August 2019 from the Funiu Mountains in Henan Province, China. The plants were identified and authenticated by *prof.* Cheng-ming Dong of the Henan University of Chinese Medicine. A voucher specimen (YJC-201908) was deposited in the Engineering and Technology Center for Chinese Medicine Development of Henan Province, Zhengzhou, China.

### 3.3. Extraction and Isolation

The air-dried *Oenothera biennis* L. (40.0 kg) was subjected to extraction with an aqueous solution containing 50% acetone of tissue fragmentation (two times each 85 L, overnight). The extract (3.3 kg) was dispersed in H_2_O (15 L) and sequentially extracted with PE (10 × 15 L), CH_2_Cl_2_(10 × 15 L), EtOAc (10 × 15 L), and *n*-BuOH (10 × 15 L). The CH_2_Cl_2_ part (49.6 g) was separated using silica gel column chromatography (CC, 12 × 130 cm) involving a mobile phase of PE/EtOAc (100:0 to 0:100, *v*/*v*), yielding 10 fractions (Fr.1–Fr.10).

Subfraction Fr.8 (2.47 g) was fractionally eluted via silica gel column containing PE/EtOAc (20:1 to 0:1, *v*/*v*), which yielded nine subfractions (Fr.8-1–Fr.8-9). SubfractionFr.8-6 (536.0 mg) was repeatedly separated via silica gel column containing CH_2_Cl_2_/MeOH (50:1 to 10:1, *v*/*v*), yielding seven subfractions (Fr.8-6-1–Fr.8-6-7). Subfraction Fr.8-6-5 (356.5 mg) was applied to a Sephadex LH-20 column containing MeOH/H_2_O (70:30, *v*/*v*), and the product was purified by semipreparative HPLC (MeOH/H_2_O 66:34), yielding compound **5** (3.4 mg, *t_R_* = 33.2 min) and compound **12** (16.2 mg, *t*_R_ = 36.0 min). Then, the subfraction Fr.8-9 (950.0 mg) was fractionally separated via silica gel column containing CH_2_Cl_2_/MeOH (100:1 to 10:1, *v*/*v*), yielding four subfractions (Fr.8-9-1–Fr.8-9-4).Then, subfraction Fr.8-9-3 (251.6 mg) was fractionally separated via silica gel column containing CH_2_Cl_2_/MeOH (50:1 to 10:1, *v*/*v*) and purified using semipreparative HPLC (MeOH/H_2_O = 66:34), yielding compound **1** (20.6 mg, *t_R_* = 35.7 min) and compound **2** (11.0 mg, t_R_ = 50.4 min). The subfraction Fr.8-9-4 (341.5 mg) was chromatographed with Sephadex LH-20 MeOH/H_2_O (70:30, *v*/*v*) and purified using semipreparative HPLC (MeOH/H_2_O = 62:38), yielding compound **6** (14.7 mg, t_R_ = 29.1 min) and compound **10** (4.6 mg, *t_R_* = 34.0 min). Sequentially, the subfraction Fr.8-9-5 (208.5 mg) was separated via silica gel column containing CH_2_Cl_2_/MeOH (40:1 to 5:1, *v*/*v*) and purified using semipreparative HPLC (MeOH/H_2_O = 53:47), which yielded compound **7** (33.8 mg, *t_R_* = 33.0 min).

In addition, subfraction Fr.9 (1.58 g) was fractionally separated via silica gel column containing CH_2_Cl_2_/MeOH (100:1 to 0:100, *v*/*v*), which yielded five subfractions (Fr.9-1–Fr.9-5). Then, the subfraction Fr.9-4 (0.78 g) was separated via silica gel column containing CH_2_Cl_2_/MeOH (50:1 to 10:1, *v*/*v*), yielding four subfractions (Fr.9-4-1–Fr.9-4-4). Further separation of the subfraction Fr.9-4-3 (304.2 mg) was chromatographed with Sephadex LH-20 MeOH/H_2_O (70:30, *v*/*v*) and purified using semipreparative HPLC (MeOH/H_2_O 65:35), producing compound **4** (11.7 mg, *t_R_* = 30.6 min) and compound **3** (12.8 mg, *t_R_* = 37.5 min). Meanwhile, subfraction Fr.9-4-4 (315.2 mg) was eluted via an MCI gel CHP-20 CC containing MeOH/H_2_O (0:100 to 100:0, *v*/*v*), yielding nine fractions (Fr.9-4-4-1–Fr.9-4-4-4). Then, subfraction Fr.9-4-4-2 (194.2 mg) was separated via silica gel column containing CH_2_Cl_2_/MeOH (40:1 to 10:1, *v/v*) and purified with semipreparative HPLC (MeOH/H_2_O 70:30), yielding compound **8** (26.4 mg, *t_R_* = 31.3 min) and compound **9** (23.9 mg, *t_R_* = 40.2 min). Subsequently, the subfraction Fr.9-5 (0.45 g) was separated via silica gel column containing CH_2_Cl_2_/MeOH (50:1 to 1:1, *v/v*), which yielded four subfractions (Fr.9-5-1–Fr.9-5-4). A Sephadex LH-20 column with MeOH/H_2_O (70:30, *v/v*) was further used to separate the subfraction Fr.9-5-2 (235.1 mg), and the obtained product was purified with semipreparative HPLC (MeOH/H_2_O = 48:52), which yielded compound **11** (12.8 mg, *t_R_* = 28.1 min).

*Oenotheralanosterol C* (**1**): The white amorphous powder of compound **1** was characterized with [α${]}_{D}^{20}$ − 71.397 (*c* 0.4120, CH_3_OH), mp 217 °C, HR-ESI-MS *m*/*z*: 511.3397 [M + Na]^+^ (calculated for C_30_H_48_O_5_Na, 511.3399) UV *λ*_max_(CH_3_OH)/nm (logε): 206 nm; IR (CH_3_OH)*v*_max_: 3369, 1636, 1457, 1019, and 682 cm^−1^. ^1^H NMR (500 MHz, CD_3_OD) data can be found in Table 1 and ^13^C NMR (125 MHz, CD_3_OD) data can be found in Table 2.

*Oenotheralanosterol**D* (**2**): The white amorphous powder of compound 2 was characterized with [α${]}_{D}^{20}$ + 1 00.290 (*c* 0.2192, CH_3_OH), mp 238 °C, HR-ESI-MS *m*/*z*: 493.2934 [M + Na]^+^ (calculated for C_29_H_42_O_5_Na, 493.2930) UV *λ*max (CH_3_OH)/nm (logε): 278, 204 nm; IR (CH_3_OH) *v*_max_: 3380, 1644, 1457, 1397, 1018, and 699 cm^−1^. ^1^H NMR (500 MHz, CD_3_OD) datacan be found in Table 1 and ^13^C NMR (125 MHz, CD_3_OD) data can be found in Table 2.

*Oenotheralanosterol**E* (**3**): The white amorphous powder of compound 3 was characterized with [α${]}_{D}^{20}$ + 69.888 (*c* 0.2336, CH_3_OH), mp 247 °C, HR-ESI-MS *m*/*z*: 509.3237 [M+Na]^+^ (calculated for C_30_H_46_O_5_Na, 509.3243) UV *λ*_max_(CH_3_OH)/nm (logε): 204 nm; IR (CH_3_OH)*v*_max_: 3368, 2941, 1686, 1459, 1383, 1238, 1018, 656, 603, and 557 cm^−1^.^1^H NMR (500 MHz, CD_3_OD) data can be found in Table 1 and ^13^C NMR (125 MHz, CD_3_OD) data can be found in Table 2.

*Oenotheralanosterol**F* (**4**): The white amorphous powder of compound 4 was characterized with [α${]}_{D}^{20}$ + 62.971 (*c* 0.2552, CH_3_OH), mp 246 °C, HR-ESI-MS *m*/*z*: 525.3180 [M+Na]^+^ (calculated for C_30_H_46_O_6_Na, 525.3192) UV *λ*_max_(CH_3_OH)/nm(logε): 203 nm; IR (CH_3_OH)*v*_max_:3366, 2939, 1688, 1455,1385, 1239, 1021,and 655 cm^−1^.^1^H NMR (500 MHz, CD_3_OD) data can be found in Table 1 and ^13^C NMR (125 MHz, CD_3_OD) data can be found in Table 2.

*Oenotheralanosterol**G* (**5**): The white amorphous powder of compound 4 was characterized with [α${]}_{D}^{20}$ + 62.971 (*c* 0.2552, CH_3_OH), mp 219 °C, HR-ESI-MS *m*/*z*: 513.2832 [M+Na]^+^ (calculated for C_28_H_42_O_7_Na, 513.2828) UV *λ*_max_(CH_3_OH) /nm(logε): 204 nm; IR (CH_3_OH)*v*_max_: 3371, 2951, 1650, 1388, 1202,1018, 669,and 554 cm^−1^.^1^H NMR (500 MHz, CD_3_OD) data can be found in Table 1 and ^13^C NMR (125 MHz, CD_3_OD) data can be found in Table 2.

### 3.4. In Vitro Cell Experiment

In the previous stage, the research group conducted relevant studies on the anti-pulmonary fibrosis activities of the total extracts and some monomers of *Oenothera biennis* L. Based on this, the RTCA method was used to detect the effect of monomer compounds on TGF-*J*1-induced BEAS-2B cell damage to explore its anti-pulmonary fibrosis active ingredients.

An xCELLigence instrument (Acea Biosciences, Inc., San Diego, CA, USA) was used for the real-time cell analysis (RTCA) assay. BEAS-2B cells were plated in 16-well plates (2.5 × 10^4^ cells/well) for 24 h at 37 °C in a humidified atmosphere of 5% CO_2_. Then, these compounds, or pirfenidone at various concentrations (0.1, 1, 10, 50, and 100 μM), were added to the standard medium of TGF-*J*1 (1 ng/mL) and incubated for 24 h. Each experiment was repeated four times to obtain the mean values. Finally, the EC_50_ values of these compounds were calculated by GraphPadSigmoidal dose-response.

## 4. Conclusions

In conclusion, we successfully isolated five new triterpenoids and seven known compounds from *Oenothera biennis* L. The bioassay indicated that compounds **1**–**2**, **6**, **8**, and **11** exhibited significant anti-pulmonary fibrosis activities to TGF-*J*1-induced BEAS-2B cells, with EC_50_ values ranging from 4.7 μM to 9.9 μM. These results provide a certain theoretical basis for the further development and utilization of *Oenothera biennis* L.

## Figures and Tables

**Figure 1 molecules-27-04870-f001:**
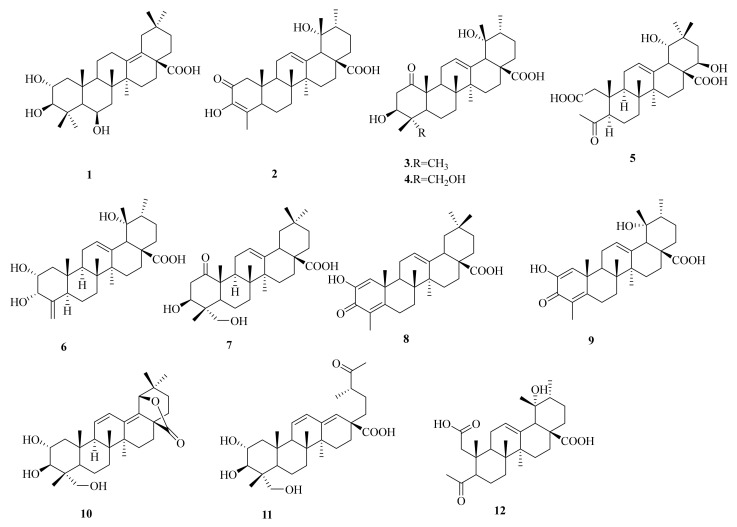
Structures of triterpenoids **1**–**12**.

**Figure 2 molecules-27-04870-f002:**
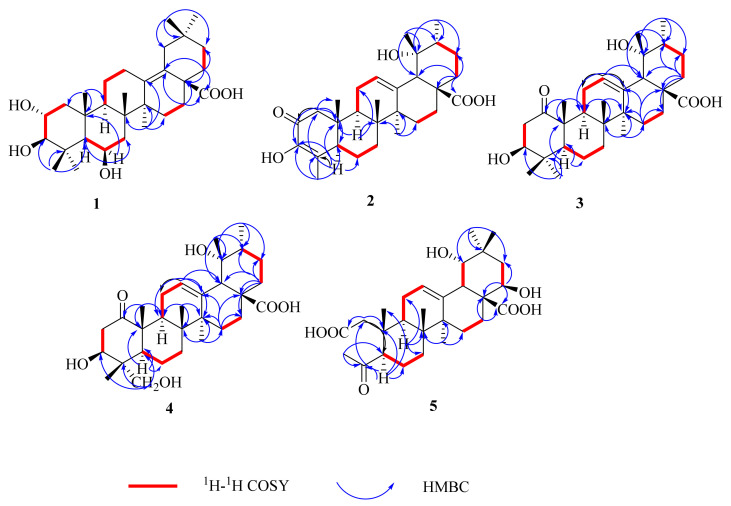
Key ^1^H-^1^H COSY and HMBC correlations of compounds **1**–**5**.

**Figure 3 molecules-27-04870-f003:**
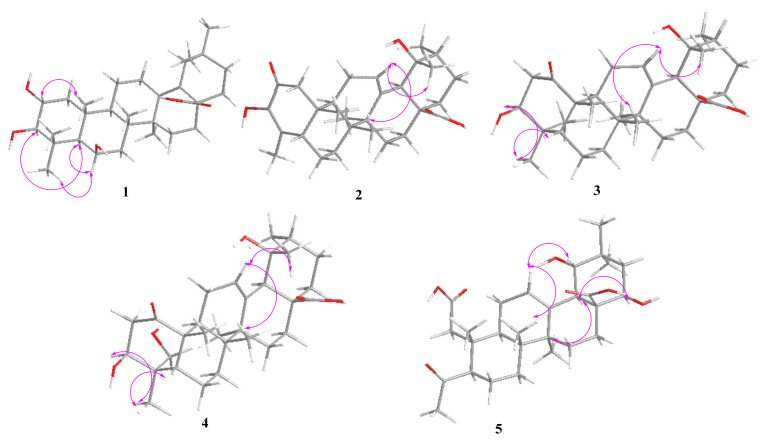
Key NOESY correlations of compounds **1**–**5**.

**Table 1 molecules-27-04870-t001:** H NMR data (500 MHz) for compounds **1**–**5** (*δ,* ppm, *J*, Hz) in CD_3_OD.

Position	1	2	3	4	5
1	2.04*dd* (4.8, 12.4), 0.90 m	2.56*d* (16.4)	-	-	2.36 s, 2.22 s
2	3.70 m	-	3.10 t (4.6), 2.26*dd*, (4.6, 12.1)	3.11 m, 2.30 m	-
3	2.86*d* (9.5)	-	3.36*dd* (4.6, 12.1)	3.82*dd* (4.9, 12.1)	-
4	-	-	-	-	-
5	0.83*d* (2.0)	2.42 m	0.90 m	1.40 m	3.06*dd* (3.0, 12.4)
6	4.45 m	1.90 m, 1.48 m	1.68 m, 1.61 m	1.60 m, 1.53 m	1.76 m, 1.67 m
7	1.64 m, 1.63 m	1.73 m, 1.43 m	1.51 m, 1.39 m	1.59 m, 1.30 m	1.59 m, 1.36 m
8	-	-	-	-	-
9	1.62 m	2.08 m	2.34 m	2.39 m	2.33 m
10	-	-	-	-	-
11	1.61 m, 1.52 m	2.07 m, 2.02 m	1.88 m, 1.54 m	2.46 m, 1.90 m	2.19 m, 2.04 m
12	2.82 m, 1.88 m	5.28 t (3.8)	5.27 t (3.6)	5.27 t (3.0)	5.35 t (3.8)
13	-	-	-	-	-
14	-	-	-	-	-
15	1.14 m, 1.06 m	1.81 m, 1.06 m	1.82 m, 1.76 m	1.78 m, 1.02 m	1.70 m, 1.11 m
16	1.91 m, 1.55 m	2.71 m, 1.57 m	2.57 m, 2.44 m	2.57 m, 2.45 m	2.00 m, 1.97 m
17	-	-	-	-	-
18	-	2.76 s	2.50 s	2.50 s	2.76 brs
19	2.47*dd*, 1.76*dd* (2.2, 14.0)	-	-	-	3.22*d* (3.8)
20	-	1.69 m	1.34 m	1.32 m	-
21	1.28 m, 1.23 m	2.31 m, 1.19 m	1.75 m, 1.26 m	1.73 m, 1.25 m	1.80 m, 1.22 m
22 23	2.15 m, 1.34 m 1.07 s	1.84 m, 1.55 m 1.85*d* (2.0)	1.77 m, 1.63 m 1.01 s	1.72 m, 1.63 m 3.35*d*, 3.51*d* (11.2)	3.91*dd* (4.4, 11.6) 2.23 s
24	1.17 s	-	1.04 s	0.88 s	-
25	1.32 s	0.93 s	1.32 s	1.35 s	1.09 s
26	1.16 s	0.85 s	0.86 s	0.86 s	0.84 s
27	1.17 s	1.39 s	1.36 s	1.37 s	1.34 s
28	-	-	-	-	-
29	0.93 s	1.14 s	1.22 s	1.22 s	0.97 s
30	0.77 s	1.00*d* (6.6)	0.94*d* (6.7)	0.94*d* (6.6)	1.03 s

**Table 2 molecules-27-04870-t002:** **^13^**C NMR data (125 MHz) for compounds **1**–**5** (*δ,* ppm, *J*, Hz) in CD_3_OD.

Position	1	2	3	4	5
1	50.5	53.2	215.3	215.3	44.4
2	69.8	195.7	45.1	44.8	175.4
3	84.7	145.3	79.7	73.4	-
4	41.1	133.1	40.4	44.1	215.6
5	56.8	49.9	55.8	47.7	57.5
6	68.8	22.0	19.0	18.6	23.1
7	43.1	33.2	34.1	33.6	32.2
8	41.6	40.6	40.9	40.8	43.3
9	52.6	44.9	40.1	40.0	40.8
10	39.3	42.3	53.8	53.3	40.6
11	23.0	24.8	26.6	26.6	24.6
12	26.5	128.1	130.0	130.0	125.1
13	139.1	140.0	139.4	139.4	143.8
14	45.9	42.9	42.8	42.9	40.6
15	28.2	29.5	29.6	29.6	29.0
16	34.2	27.2	26.5	26.5	20.8
17	49.4	49.0	49.3	49.3	53.0
18	129.8	48.0	55.3	55.3	46.4
19	42.0	74.2	73.5	73.5	82.0
20	33.6	43.3	43.1	43.1	36.8
21	38.0	25.2	27.3	27.3	38.0
22	36.9	32.8	38.9	38.9	72.4
23	28.8	13.4	16.6	66.0	31.6
24	18.7	-	29.0	13.3	-
25	19.5	14.4	15.4	15.9	18.4
26	19.8	17.7	18.1	18.1	17.5
27	21.8	24.5	24.8	24.8	25.2
28	180.6	182.2	182.2	182.2	180.7
29	32.7	29.5	27.0	27.0	28.8
30	24.7	16.2	16.6	16.6	26.2

**Table 3 molecules-27-04870-t003:** Anti-pulmonary fibrosis activities of compounds **1**–**12** against BEAS-2B cell damage induced by TGF-*J*1.

Group	EC_50_ (μM)	Group	EC_50_ (μM)
CON	NA	**6**	4.7
M	NA	**7**	53.3
Pirfenidone	12.7	**8**	7.9
**1**	8.4	**9**	NA
**2**	9.9	**10**	NA
**3**	48.3	**11**	9.6
**4**	NA	**12**	NA
**5**	NA		

Note: CON: Normal group; M: Model group; Pirfenidone: Positive control compound.

## Data Availability

The data presented in this study are available on request from the corresponding author.

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
