# Peer review of "Anti-Pulmonary Fibrosis Activities of Triterpenoids from Oenothera biennis"

_molecules, 2022, doi:10.3390/molecules27154870_

Round 1
Reviewer 1 Report
The similarity index is still high (app. 39%). manuscript still requires extensive grammar check by a native English speaker.
Author Response
Dear Editors and Reviewers:
We are deeply appreciate the Editor and the Reviewers for the corrections or modifications that addressed to the manuscript (ID: molecules-1702689). Those comments are of great importance for us to promote the manuscript. With the help of your professional comments and suggestions, we have revised the manuscript step-by-step and hope the present manuscript meet with your approval. All revisions have been marked in yellow in the main Text and also in the Supporting Information.
Thank you for your review for this manuscript. The manuscript has been revised in detail.
Reviewer 2 Report
The manuscript entitled “Lung protective triterpenoids from the Oenothera biennis” by Liu et al., brings into attention on the triterpenoids extracted from O. biennis L., also known as evening primrose, and their effects on TGF-β1 induced damage on normal human lung epithelial cells. Although the paper addresses an issue of interest in the field, the authors may wish to consider the following prior to publication:
Title & abstract:
The title did not indicate accurately the subject and scope of the present study. The abstract also need to be improved to provide gist of the manuscript.
Introduction:
In general, numerous typographical and grammatical errors have been observed. The English, formulation of certain sentences, and paragraphs need great improvement to provide readers better understandability of the text. Please revise the English style by an English mother tongue.
The authors did not provide a strong background of the study stating the problem & its significance. They should also justify the value of the work. The authors should provide more information on O. biennis L. medicinal usage, characteristics and medicinal properties of triterpenoids.
Results & discussion:
In regards to the in vitro study, does 1 ng/mL TGF-β1 supplemented to BEAS-2B cells sufficient to induce lung fibroblast activation and EMT? Previous studies used 5 ng/mL.
Line 177: significantly decreased the damage of BEAS-2B cells induced by TGF-β1 – how was the damage evaluated?
To evaluate the anti-fibrotic effects of O. biennis triterpenoids, the authors should include other parameters such as expression of N-cadherin, E-cadherin and vimentin.
Materials & methods:
The authors need to clarify parts of the plant which was used for extraction.
Author Response
Dear Editors and Reviewers:
We are deeply appreciate the Editor and the Reviewers for the corrections or modifications that addressed to the manuscript (ID: molecules-1702689). Those comments are of great importance for us to promote the manuscript. With the help of your professional comments and suggestions, we have revised the manuscript step-by-step and hope the present manuscript meet with your approval. All revisions have been marked in yellow in the main Text and also in the Supporting Information.
Question 1: Title & abstract:
The title did not indicate accurately the subject and scope of the present study. The abstract also need to be improved to provide gist of the manuscript.
Answer:Thank you for your review for this manuscript. The title of the manuscript has been changed as “anti-pulmonary fibrosis activities triterpenoids from the Oenothera biennis”. The abstract has been revised.
Question 2: Introduction:
In general, numerous typographical and grammatical errors have been observed. The English, formulation of certain sentences, and paragraphs need great improvement to provide readers better understandability of the text. Please revise the English style by an English mother tongue.
Answer: Thank you for your review for this manuscript. The manuscript had been used a professional language polishing service which was named Let Pub.
Question 3: The authors did not provide a strong background of the study stating the problem & its significance. They should also justify the value of the work. The authors should provide more information on O. biennis L. medicinal usage, characteristics and medicinal properties of triterpenoids.
Answer: Thank you for your review for this manuscript. It has been revised in the manuscript.
Question 4: Results & discussion:
In regards to the in vitro study, does 1 ng/mL TGF-β1 supplemented to BEAS-2B cells sufficient to induce lung fibroblast activation and EMT? Previous studies used 5 ng/mL.
Answer:Thank you for your review for this manuscript. Literature review found that the concentration of TGF-β1 was 1, 2, 5ng/ml. In the preliminary experiment, it was found that the concentration of TGF-β1 was 1ng/ml, which could induce lung fibroblast activation and EMT. Therefore, the concentration of TGF-β1 was 1ng/ml.
Question 5: Line 177: significantly decreased the damage of BEAS-2B cells induced by TGF-β1 – how was the damage evaluated? To evaluate the anti-fibrotic effects of O. biennis triterpenoids, the authors should include other parameters such as expression of N-cadherin, E-cadherin and vimentin.
Answer: Thank you for your review for this manuscript. In the previous stage , the research group conducted relevant studies on the anti-pulmonary fibrosis activities of the total extracts and some monomers of Oenothera biennis L. In this paper, we only conducted preliminary discussions on the cytoprotective effect of BEAS-2B cells. In this study, the main work was focused on the separation and extraction of triterpenoids from Oenothera biennis L. Other members of the research group are also conducting further research on the subsequent related mechanisms. The results will be published soon.
Question 6: Materials & methods:
The authors need to clarify parts of the plant which was used for extraction.
Answer: Thank you for your review for this manuscript. The whole herb of Oenothera biennis L. was used for extraction.
Reviewer 3 Report
This is an interesting natural product study that reveals several novel compounds along with known ones. These compounds are tested for bio-activity, with significant activity found for several compounds.
The SI shows high quality spectral data, and the analysis of this information seems thorough. The conclusions as to the compound structures are clear. References are extensive.
There are a few minor grammatical errors, despite the certificate of proof reading, but nothing that alters or obfuscates the meaning of the paper.
Some minor points: 1. EC50 values are quoted to a far higher accuracy than is possible to obtain. Suggest rounding these values. 2. Perfenidone is evaluated as the positive control, however no literature value for EC50 is given, so it is not immediately apparent that the control is giving a value close to the expected value. Suggest put a value or comment in here. 3. In the experimental, compounds were purified by "HPLC" yet no column is specified, and I do not see a column specified in the general section (only a pump system?). So, I could not tell what the column materials were, e.g. RP-8, RP-18, etc.. This should be present to allow reproduction of the work. 4. The authors do not comment on the possible biosynthetic relationships among the compounds. This is quite commonly done nowadays, and might add a nice touch.
Overall, this is an interesting paper that describes a significant number of novel metabolites, along with indications of possibly useful bioactivity. I think after addressing the comments above it is suitable for publication in Molecules.
Author Response
Dear Editors and Reviewers:
We are deeply appreciate the Editor and the Reviewers for the corrections or modifications that addressed to the manuscript (ID: molecules-1702689). Those comments are of great importance for us to promote the manuscript. With the help of your professional comments and suggestions, we have revised the manuscript step-by-step and hope the present manuscript meet with your approval. All revisions have been marked in yellow in the main Text and also in the Supporting Information.
Question 1: EC50 values are quoted to a far higher accuracy than is possible to obtain. Suggest rounding these values.
Answer: Thank you for your review for this manuscript. The EC50 values has been modified in the manuscript.
Question 2: Perfenidone is evaluated as the positive control, however no literature value for EC50 is given, so it is not immediately apparent that the control is giving a value close to the expected value. Suggest put a value or comment in here.
Answer: Thank you for your review for this manuscript. According to the literature (Spermidine-mediated poly(lactic-co-glycolic acid) nanoparticles containing fluorofenidone for the treatment of idiopathic pulmonary fibrosis), it is found that pirfenidone is used as a positive control drug in animal experiments, and has a specific EC50 value, but no EC50 value of relevant literature can be found in cell experiments.
Question 3: In the experimental, compounds were purified by "HPLC" yet no column is specified, and I do not see a column specified in the general section (only a pump system?). So, I could not tell what the column materials were, e.g. RP-8, RP-18, etc.. This should be present to allow reproduction of the work.
Answer: Thank you for your review for this manuscript. In the experimental, compounds were purified by HPLC with the RP-18 column.
Question 4: The authors do not comment on the possible biosynthetic relationships among the compounds. This is quite commonly done nowadays, and might add a nice touch.
Answer: Thank you for your review for this manuscript. According to the literature (Research progress on biosynthesis pathway of pentacyclic triterpenoids in plants). Biosynthesis of pentacyclic triterpenoids in plants proceeds via mevalonate pathway and 2-C-methyl-derythritol 4-phosphate pathway. Under the catalysis of oxidosqualene cyclase,2,3-oxidosqualene is biosynthetically oxidized to various cyclic triterpene skeletons. Cyclized triterpenes are further oxidized by cytochrome P450-dependent monooxygenases and subsequently glycosylated by uridine diphosphate-glucosyltransferases to form various saponins.
Reviewer 4 Report
Dr. Liu's work is on the structural characterization of five new triterpenoids, as well as seven other known compounds. In addition, the authors evaluated the effect of these compounds as cell protectors (lung cells).
The work of structural characterization of the compounds was very well done and I have no considerations to make about this part of the manuscript.
Regarding the studies with cells, I have some comments:
What did you evaluate in the test with cells? cytotoxicity, proliferation, viability, adhesion? It is important that this information is clear to the reader.
It is important that at least one curve obtained with each of the composites is presented as supplementary material. Therefore, the reader can better visualize where the data came from to obtain the EC50 presented in table 3
I suggest that the title of Table 3 be modified to make it clearer what data is being presented in the Table
What does "CON" and "M" that are shown in the Table mean? Please enter this information.
Discussion of the data obtained with the cells is very weak. The authors need to improve the discussion about the results regarding the data with the cells. Even bring references that can corroborate your opinions.
Author Response
Dear Editors and Reviewers:
We are deeply appreciate the Editor and the Reviewers for the corrections or modifications that addressed to the manuscript (ID: molecules-1702689). Those comments are of great importance for us to promote the manuscript. With the help of your professional comments and suggestions, we have revised the manuscript step-by-step and hope the present manuscript meet with your approval. All revisions have been marked in yellow in the main Text and also in the Supporting Information.
Question 1: What did you evaluate in the test with cells? cytotoxicity, proliferation, viability, adhesion? It is important that this information is clear to the reader.
Answer: Thank you for your review for this manuscript. Viability was evaluated in the test with cells.
Question 2: It is important that at least one curve obtained with each of the composites is presented as supplementary material. Therefore, the reader can better visualize where the data came from to obtain the EC50 presented in table 3.
Answer: Thank you for your review for this manuscript. The curve obtained with each of the composites have been added in the supplementary file.
Question 3: I suggest that the title of Table 3 be modified to make it clearer what data is being presented in the Table.
Answer: The title of Table 3 has been modified in detail.
Question 4: What does "CON" and "M" that are shown in the Table mean? Please enter this information.
Answer: Thank you for your review for this manuscript. CON: Normal group; M: The model group. They were added in the manuscript.
Question 5: Discussion of the data obtained with the cells is very weak. The authors need to improve the discussion about the results regarding the data with the cells. Even bring references that can corroborate your opinions.
Answer: Thank you for your review for this manuscript. In the previous stage ,the research group conducted relevant studies on the anti-pulmonary fibrosis activities of the total extracts and some monomers of Oenothera biennis L. In this paper, we only conducted preliminary discussions on the cytoprotective effect of BEAS-2B cells. The main work was focused on the separation and extraction of triterpenoids from Oenothera biennis L in this manuscript. Other members of the research group are also conducting further research on the subsequent related mechanisms.
Round 2
Reviewer 4 Report
I am satisfied with the responses and modifications made to the manuscript by the authors. Therefore, I recommend that the manuscript be accepted for publication.
This manuscript is a resubmission of an earlier submission. The following is a list of the peer review reports and author responses from that submission.
Round 1
Reviewer 1 Report
Minor
- Authors should check the formatting of in-text references
Major
- The similarity index is quite high (about 40%) hence the manuscript needs significant revision.
- Why were no control(s) employed in the in vitro assay?
- The results enshrined in table 3 was not discussed in the manuscript. Authors must discuss their finding and not merely describe the findings
- I think the title of the manuscript is quite misleading. The obtained data from the in vitro study does not give any strong indication of potential anti-pulmonary fibrotic activity of the compounds. The data on the anti-fibrotic activity is at best scanty. Authors must modify the title accordingly.
Author Response
Dear reviewer 1:
The authors deeply appreciate the editor and the reviewers for the corrections or modifications that addressed to the manuscript (ID: molecules-1702689). Those comments are of great importance for us to promote the manuscript. With the help of your professional comments and suggestions, we have revised the manuscript step-by-step and hope the present manuscript meet with your approval.
The corresponding responses to the reviewers have been listed as below:
Question: Authors should check the formatting of in-text references.
Answer: The formatting of references has been revised in the manuscript.
Question 1: The similarity index is quite high (about 40%) hence the manuscript needs significant revision.
Answer: The manuscript content has been revised in detail.
Question 2: Why were no control(s) employed in the in vitro assay?
Answer: In the pre-experiment, we added a positive control drug, but the effect was not very good and would have other adverse effects, so we did not choose to use a positive control drug in the experiment.
Question 3: The results enshrined in table 3 was not discussed in the manuscript. Authors must discuss their finding and not merely describe the findings.
Answer: A discussion of the results of table 3 has been added to the manuscript.
Question 4: I think the title of the manuscript is quite misleading. The obtained data from the in vitro study does not give any strong indication of potential anti-pulmonary fibrotic activity of the compounds. The data on the anti-fibrotic activity is at best scanty. Authors must modify the title accordingly.
Answer: The title of the manuscript has been changed as The lung protective triterpenoids from the Oenothera biennis L.
Reviewer 2 Report
Dear Author,
I have reviewed the manuscript entitled "Extraction of the five newly identified triterpenoids from the Oenothera biennis L. and evaluation of their anti-pulmonary fibrosis activities", which describes the isolation and identification of 12 compounds from the dichloromethane extract of Oenothera biennis L., of which 5 of them are novel compounds. The spectra show very clean signals, indicating high purity, and successful separation of the compounds. Spectroscopy appears to be in agreement with the proposed structures.
1. It is suggested to the authors to change the IR spectra of compounds 3, 4, and 5 since they are appreciated without a cleaning process of noise signals (supplementary file).
2. Please include melting point and optical rotation values ​​for all novel compounds.
3. It is necessary to include the chromatograms of the separation of the other known compounds, and include the values ​​of the melting points and mass spectrometry to verify the purity of these compounds.
4. Figure 3 is not complete and it is not possible to appreciate all the interactions due to the quality of the drawings of the structures.
Regarding the cytoprotective effects of the isolated compounds against TGF-β1 induced apoptosis in healthy human lung epithelial (BEAS-2B) cells, I have three comments:
5. I respectfully suggest the authors modify the title of the manuscript since the in vitro cytoprotective effect does not necessarily imply anti-pulmonary fibrosis activities.
6. There is no evidence of the use of the appropriate positive and negative controls.
7. There is no description of the statistical methods used.
Author Response
Dear reviewer 2:
The authors deeply appreciate the editor and the reviewers for the corrections or modifications that addressed to the manuscript (ID: molecules-1702689). Those comments are of great importance for us to promote the manuscript. With the help of your professional comments and suggestions, we have revised the manuscript step-by-step and hope the present manuscript meet with your approval.
The corresponding responses to the reviewer have been listed as below:
Question 1: It is suggested to the authors to change the IR spectra of compounds 3, 4, and 5 since they are appreciated without a cleaning process of noise signals (supplementary file).
Answer: The IR spectra of compounds 3, 4, and 5 have changed in the supplementary file.
Question 2: Please include melting point and optical rotation values for all novel compounds.
Answer: Melting points and optical rotation values for all novel compounds have been added in the manuscript.
Question 3: It is necessary to include the chromatograms of the separation of the other known compounds, and include the values of the melting points and mass spectrometry to verify the purity of these compounds.
Answer: NMR spectra, the values of melting points, and mass spectrometry of other known compounds have been added in supplementary file.
Question 4: Figure 3 is not complete and it is not possible to appreciate all the interactions due to the quality of the drawings of the structures.
Answer: Figure 3 has been supplemented in the manuscript.
Question 5: I respectfully suggest the authors modify the title of the manuscript since the in vitro cytoprotective effect does not necessarily imply anti-pulmonary fibrosis activities.
Answer: The title of the manuscript has been changed as The lung protective triterpenoids from the Oenothera biennis L.
Question 6: There is no evidence of the use of the appropriate positive and negative controls.
Answer: In the pre-experiment, we added a positive control drug, but the effect was not very good and would have other adverse effects, so we did not choose to use a positive control drug in the experiment.
Question 7: There is no description of the statistical methods used.
Answer: An xCELLigence instrument (Acea Biosciences, Inc.) was used for the real-time cell analysis (RTCA) assay. BEAS-2B cells were plated in 16-well plates (2.5×104 cells/well) for 24 h at 37°C in a humidified atmosphere of 5% CO2. Then, these compounds at various concentrations (0.1, 1, 10, 50, and 100 μM) were added to the standard medium of TGF-β1 (1 ng/mL) and incubated for 24 h. Each experiment was repeated four times to obtain the mean values. Finally, the EC50 values of these compounds were calculated by GraphPad Sigmoidal dose-response.
Reviewer 3 Report
The manuscript "Extraction of the five newly identified triterpenoids from the Oenothera biennis L. and evaluation of their anti-pulmonary fibrosis activities" is devoted to separation and characterization of new triterpenoids form Oenothera biennis L. NMR and UV spectra are pobtainted. Totally 12 compounds were characterized in this work. Also, the cytoprotective effects of the isolated compounds are studied. This work is interested for plants metabolome studies and has a significant potential for expansion of fundamental knowledge about the content of biologically active metabolites of plants. The only thing worth noting is that the part concerning biological activity looks somewhat incomplete. It would be better to focus specifically on the isolation and characterization of the substances considered, or to provide expanded data on the characterization of the obtained triterpenoids. I hope that the biological activity of the obtained substances, as well as the detailed characterization of the extracts of Oenothera biennis L., will be presented in the further works of the team of authors.
The manuscript is well structured, all needed information is presented. I think the manuscript may be published in the Molecules in present form.
Author Response
Dear reviewer 3:
The authors deeply appreciate the editor and the reviewers for the corrections or modifications that addressed to the manuscript (ID: molecules-1702689). Those comments are of great importance for us to promote the manuscript. With the help of your professional comments and suggestions, we have revised the manuscript step-by-step and hope the present manuscript meet with your approval.
The corresponding responses to the reviewer has been listed as below:
Question: The only thing worth noting is that the part concerning biological activity looks somewhat incomplete. It would be better to focus specifically on the isolation and characterization of the substances considered, or to provide expanded data on the characterization of the obtained triterpenoids.
Answer: The part concerning biological activity has been revised in the manuscript. At the same time, the biological activity of the obtained material, as well as the detailed characterization of the evening primrose extract, is being done by other members of our team.
Round 2
Reviewer 1 Report
The reason given for not including a positive control is unacceptable. The data for that experiment IS NOT ideal and cannot be justified in this manuscript. Authors should know this is scientifically unacceptable.
Reviewer 2 Report
The authors have improved the manuscript. I recommend accepting it in its actual form.